# In Vitro Metabolism and CYP-Modulating Activity of Lavender Oil and Its Major Constituents

**DOI:** 10.3390/molecules28020755

**Published:** 2023-01-12

**Authors:** Goutam Mondal, Olivia R. Dale, Yan-Hong Wang, Shabana I. Khan, Ikhlas A. Khan, Charles R. Yates

**Affiliations:** 1National Center for Natural Products Research, School of Pharmacy, University of Mississippi, University, MS 38677, USA; 2Division of Pharmacognosy, Department of BioMolecular Sciences, School of Pharmacy, University of Mississippi, University, MS 38677, USA

**Keywords:** linalool, metabolism, essential oil

## Abstract

The application of essential oils has historically been limited to topical (massage therapy) and inhalational (aromatherapy) routes of administration. More recently, however, evaluation of the therapeutic effects of essential oils has expanded to include the oral route of administration, which increases the herb–drug interaction potential. The purpose of this study was to evaluate the herb–drug interaction potential of lavender essential oil and two of its primary phytoactive constituents, namely linalool and linalyl acetate. The metabolic stability of linalool and linalyl acetate was determined in human liver microsomes (HLM) and S9 fractions by quantitative analysis using UPLC-MS/MS system. Linalool was metabolically unstable in HLM and S9 fractions with an intrinsic clearance of 31.28 mL·min^−1^·kg^−1^, and 7.64 mL·min^−1^·kg^−1^, respectively. Interestingly, it was observed that linalyl acetate converted to linalool both in HLM and S9 fractions. Lavender oil showed weak inhibitory effect on the catalytic activity of CYP3A4 and CYP1A2 enzymes (IC_50_ 12.0 and 21.5 µg/mL). Linalyl acetate inhibited CYP3A4 (IC_50_ 4.75 µg/mL) while linalool did not show any inhibitory effect on any of the enzymes. The lavender oil and its constituents did not activate PXR to a considerable extent, and no activation of AhR was observed, suggesting a lack of potential to modify the pharmacokinetic and pharmacodynamic properties of conventional medications if used concurrently.

## 1. Introduction

Essential oils (EOs) and their constituents have been used in aromatic and topical therapeutic modalities from ancient times and are purported to possess anxiolytic, sedative, and calming properties [1,2]. However, more recently, EOs and their constituents have been increasingly recommended for oral administration [2]. For example, Silexan is a standardized essential oil (SLO) extract of *Lavandula angustifolia*, which has been developed and approved for oral use in subsyndromal anxiety [3,4]. Numerous clinical trials have demonstrated the effectiveness of orally administered Silexan against typical anxiety-related sequelae including impaired sleep, depression, decreased quality of life, and various somatic complaints [5]. In addition, the dietary supplement CalmAid^®^ (SLO 80 mg) is available in the United States for once- or twice-daily oral administration carrying structure function claims for tension and stress [1]. However, no peer-reviewed clinical studies have been conducted with CalmAid.

Because EOs have been infrequently administered orally, there are relatively few studies designed to gauge the drug interaction potential of EO and/or their phytoactive constituents. Samojlik et al. investigated the effect of acute and chronic oral administration of a peppermint oil (PO, *Mentha* × *piperita Lamiaceae*) emulsion on the pharmacologic effects of pentobarbitone (sleeping time), codeine (analgesic effect), and midazolam (impaired motor coordination) in mice [6]. The authors noted that acute PO administration prolonged pentobarbitone-induced sleep time without affecting codeine-related analgesia or midazolam-related motor coordination impairment. In contrast, chronic oral administration of PO decreased codeine’s analgesic effects of codeine, shortened pentobarbitone-induced sleep time, and both enhanced and prolonged midazolam’s motor coordination effects. In a separate study, Samojlik et al. investigated the impact of orally administered aniseed (*Pimpinella anisum*) EO (0.3 mg/kg/day × 5 days) on the oral and intraperitoneal pharmacokinetics of acetaminophen (200 mg/kg) and caffeine (20 mg/kg) in mice [7]. Aniseed EO reduced acetaminophen’s peak plasma concentration and total area under the plasma curve (AUC). In addition, aniseed EO significantly reduced the bioavailability of caffeine.

Relatively few human clinical drug interaction studies have been conducted and these have been limited to ones examining LEO’s interaction potential. For example, Doroshyenko et al. evaluated LEO’s (Silexan, 160 mg once daily × 11 days) interaction potential using a drug cocktail comprised of substrates for CYP1A2 (caffeine, 150 mg), CYP2C9 (tolbutamide, 125 mg), CYP2C19 (omeprazole, 20 mg), CYP2D6 (dextromethorphan, 30 mg), and CYP3A4 (midazolam, 2 mg), which was administered orally on study day 12 [8]. LEO administration had no impact on the primary measures of substrate pharmacokinetics, viz., maximum concentration, half-life, time to maximum concentration, or AUC. Thus, the authors concluded that LEO, at the clinically relevant dose, posed little risk of herb–drug interaction. In a separate double-blind, randomized, two-period crossover study, the effect of simultaneous oral administration of LEO (Silexan, 160 mg once daily × 21 days) on the pharmacokinetics and pharmacodynamics of Microgynon^®^ (ethinyl estradiol 0.03 mg and levonorgestrel 0.15 mg once daily for 28 days) was investigated [9]. The maximum concentration and AUC of ethinyl estradiol and levonorgestrel, both of which are CYP3A4 substrates [10], were unaffected by concomitant oral LEO administration. It is tempting to conclude that LEO and its phytoactive constituents pose little risk for drug interactions. However, in the absence of human pharmacokinetic data, which is the case for orally administered LEO, one must interpret the aforementioned results with caution.

To gain further insight into the drug interaction potential of LEO, a few studies have examined the in vitro metabolism of linalool, one of the most abundant phytoactives in LEO. For example, linalool increased the metabolic activity of CYP2A and was a weak competitor of CYP2C6 in rat liver microsomes [11]. In another study, recombinant human cytochrome P450 enzymes were used to demonstrate CYP2C19- and CYP2D6-mediated allylic hydroxylation and epoxidation of linalool [12]. However, to our knowledge, there has been no systematic evaluation of the ability of LEO phytoactives to inhibit and/or induce cytochrome P450 enzymes. In the present study, we evaluated the inhibitory effects of LEO phytoactives towards two major drug metabolizing enzymes (CYP3A4 and 1A2) which account for the metabolism of ~54% marketed drugs, using C-DNA baculovirus-expressed recombinant enzyme isoforms and specific fluorescent substrates. Further, the nuclear receptor, PXR, is a master regulator of DME as well as the major transcriptional regulator of the CYP3A4 and CYP1A2 genes [13]. We evaluated the ability of LEO phytoactives to modulate the activity of PXR and AhR receptors using reporter gene assays carried out in PXR-transfected HepG2 cells and a proprietary AhR reporter cell line, respectively. Additionally, we determined the in vitro metabolic stability of the LEO phytoactives, linalool and linalyl acetate, and estimated their in vitro intrinsic clearance (CL_int’_) by determining the half-life (t_1/2_) and the elimination rate constant (k_el_). 

## 2. Results

### 2.1. In Vitro Metabolism

Liver metabolism is a very important factor which determines the systemic bioavailability of a compound. Phase I reactions are the addition or unmasking of functional, polar moieties by oxidation (CYP450 or FMO) or hydrolysis (esterases), whereas phase II reactions are the conjugation with small, endogenous substances carried out by the UGT. The assessment of phase I and phase II metabolism were carried out in HLM (phase I) and S9 fractions (phase I and phase II) using NADPH and UDPGA as cofactors, respectively. Prediction of in vivo drug clearance can be made from the in vitro intrinsic clearance (CL_int’_) data of a drug [14]. Measurement of in vitro microsomal half-life (t_1/2_) is the simplest approach for determining CL_int’_ (mL/min/kg) [15]. Lavender oil contains numerous active ingredients like linalool, linalyl acetate, terpinen-4-ol, and trans-cis-β-ocimene; among these active ingredients, linalool and linalyl acetate are the major active ingredients responsible for biological properties of lavender oil [16]. 

High and nearly equivalent content of linalool and linalyl acetate is essential for the biological properties of lavender essential oil [16]. We selected linalool and linalyl acetate for in vitro metabolism studies as well as for the evaluation of CYP inhibition, and PXR and AhR activation effects. Linalool was degraded in HLM with a half-life of 19.94 min and intrinsic clearance of 31.28 mL·min^−1^·kg^−1^, but degraded in the S9 fraction slowly with a half-life of 81.67 min and intrinsic clearance of 7.64 mL·min^−1^·kg^−1^ (Figure 1 and Figure 2). This indicates that linalool could be metabolized at a faster rate by phase I enzymes, which could account for the poor systemic bioavailability seen in an in vivo pharmacokinetic study in male Sprague Dawley rats [17]. 

In contrast, the linalyl acetate was found to be rapidly degraded in HLM, and disappeared within 15 min, but degraded at a slower rate in S9 fractions with a half-life of 7.34 min and intrinsic clearance of 84.97 mL·min^−1^·kg^−1^ (Figure 2). Most importantly, it was observed that linalyl acetate was converted to linalool rapidly in HLM as well as disappeared at a faster rate with a Cmax of 118.45 ng/mL and Tmax of 10 min. On the other hand, linalyl acetate was converted slowly into linalool in S9 fractions as well as disappeared slowly with a Cmax of 248.7 ng/mL, and Tmax of 30 min (Figure 3 and Figure 4). These results indicate that linalyl acetate is metabolically unstable and converted to linalool, which accounts for its rapid metabolism into linalool as observed in an in vivo pharmacokinetic study in rats [18].

### 2.2. CYP Modulating Activity

A number of isoenzymes from the CYPs family like CYP1A2, 2A6, 2B6, 2C8, 2C9, 2C19, 2D6, 2E1, 3A4, and 3C5 are involved in drug metabolism, however, CYP1A2 and CYP3A4 play a major role in the metabolism of xenobiotics and account for the metabolism of ~54% of marketed drugs [13,19]. Therefore, CYP inhibition assays were carried out to determine if lavender oil and its constituents could inhibit the two major drug metabolizing enzymes namely CYP3A4 and CYP1A2. Lavender oil was found to inhibit the catalytic activity of CYP3A4 and CYP1A2 enzymes in a dose-dependent manner and the IC50 values were 12.0 and 21.5 µg/mL, respectively (Table 1). Among the two major constituents, linalyl acetate inhibited CYP3A4 with an IC50 value of 4.75 µg/mL while no inhibition was observed by linalool (Table 1). The activity of the CYP1A2 isoform was not inhibited by any of the compounds. It is interesting to note that the extent of CYP1A2 inhibition by lavender oil was also low compared to CYP3A4 inhibition. Inhibition of CYP3A4 by lavender oil can be explained in terms of CYP3A4 inhibition by its constituents such as linalyl acetate.

To predict the drug interaction potential of lavender oil and its constituents in terms of PXR and AhR activation, reporter gene assays were carried out to measure the transcriptional activity of PXR and AhR. Activation of PXR is known to result in increased expression of CYP3A4 which is a major CYP isoform responsible for metabolizing majority of clinical drugs [20]. At the highest tested concentration of lavender oil (60 µg/mL), PXR activity was increased to about 1.7-fold compared to the vehicle-treated control. Linalyl acetate and linalool activated PXR about 1.7-fold at 30 µg/mL (Table 2). Fold induction of 2 or more in the PXR assay is considered a significant effect. The oil did not activate PXR transcriptional activity to a considerable extent (fold activation < 2-fold at the highest tested concentration) and hence it is less likely to pose a risk to interfere with the pharmacokinetics of drugs which are CYP3A4 substrates. Altogether, these results indicate that lavender oil and its tested phytoconstituents are less likely to pose or cause any drug interactions.

Aryl hydrocarbon receptor (AhR) is known to regulate the expression of the CYP1A2 which is involved in the metabolism of caffeine, theophylline, and several antidepressants [21,22]. Increased activity of CYP1A2 may lead to increased clearance of its substrate drugs. No activation of AhR was observed by the lavender oil indicating that it may not have any concern of inducing the expression of CYP1A2 which is regulated by AhR. 

Based on our observation of the inhibitory effects on CYP3A4, it can be speculated that a net effect on the enzymatic activity of CYP3A4 would be negligible. PXR-inducing effects may result in increased expression of protein but the inhibitory action may counteract the effect.

## 3. Discussion

To our knowledge, this is the first preliminary report studying the in vitro metabolism in HLM and S9 fractions, and CYP-modulating activity of lavender oil and its major constituents. Linalool and linalyl acetate were rapidly degraded in HLM, whereas linalool as well as linalyl acetate degraded at a slower rate in S9 fractions, which is indicative of poor bioavailability of these compounds. Interestingly, it was observed that linalyl acetate converted to linalool rapidly in HLM compared to S9 fractions, which is indicative of the rapid formation of linalool from linalyl acetate in vivo. Lavender oil and its constituents showed negligible inhibitory effect on the catalytic activity of CYP3A4 while no inhibition of the activity of CYP1A2 was observed by any of the compounds. Additionally, lavender oil and its constituents showed < 2-fold induction in PXR activity while no activation of AhR was observed. By implication, the concomitant use of lavender oil and its tested phytoconstituents with prescription medications are less likely to pose any risk for pharmacokinetic drug interactions. 

## 4. Materials and Methods

### 4.1. Reagents

Dulbecco’s Modified Eagle Medium (DMEM), DMEM-F12, minimal essential medium (MEM), trypsin EDTA, penicillin-streptomycin, and sodium pyruvate were purchased from GIBCO BRL (Invitrogen Corp., Grand Island, NY, USA). FBS was purchased from Hyclone Lab Inc. (Logan, UT, USA). All other chemicals were from Sigma Chem. Co., Ltd. (St. Louis, MO, USA). Human liver microsomes and S9 fractions (pooled mixed sex) were from In Vitro Technologies Inc. G-6-PDH, glucose-6-phosphate, NADP+, UDPGA, and all other chemicals were from Sigma Chem. Co., Ltd. (St. Louis, MO, USA). Linalool, linalyl acetate, and lavender essential oil were received from doTERRA (West Pleasant Grove, UT, USA). The purity of linalool and linalyl acetate was confirmed by chromatographic analysis to be ≥95%. The purity of the positive controls was ≥98%. LEO characterization was reported previously [23].

### 4.2. LC-MS/MS Analysis

The LC-MS/MS analyses were carried out on a Waters Acquity UPLC^TM^ I-class system (Waters Corp., Milford, MA, USA) coupled with a Xevo TQ-S triple quadrupole mass spectrometry detector using a Waters UPLC BEH C_18_ column (50 mm × 2.1 mm I.D., 1.8 µm). The instrument was controlled by Waters MassLynx 4.1 software. The column and sample temperatures were maintained at 40 °C and 10 °C, respectively. The mobile phase consisted of water containing 0.1 % formic acid (A) and acetonitrile with 0.1 % formic acid (B) at a flow rate of 0.37 mL/min in the following gradient elution: 0–0.8 minutes, 50% B to 70% B; 0.8–1.7 min, held at 70% B; 1.7–2.0 min, 70% B to 100% B. The analysis was followed by a one and half minute washing procedure with 100% B and a re-equilibration period of 3.5 min with the initial condition. The injection volume was 2 µL. 

The ESI MS/MS parameters were set as follows: capillary voltage, 4.0 kV; source temperature, 150 °C; desolvation temperature, 300 °C; desolvation gas flow, 800 L/h, and cone gas flow, 150 L/h. Nitrogen was used as the desolvation and cone gas. Argon (99.99% purity) was introduced as the collision gas into the collision cell at a flow rate of 0.15 mL/min. The effluent was introduced into the TQ-S mass spectrometer in positive ion mode (ESI^+^) for quantification of the analytes. Detection was obtained by Multiple Reaction Monitoring (MRM) mode. The quantification of linalool (retention time, 1.02 min) and linalyl acetate (retention time, 1.63 min) were acquired with transitions of key product ions at *m*/*z* 137.09→95.10 (dwell time 24 ms, cone voltage 39 V, and collision energy 10 eV). For the quantification of 7-hydroxy coumarin (retention time, 0.43 min) and testosterone (retention time, 0.77 min), MRM ions transitions were at *m*/*z* 162.94→107.07 (dwell time 15 ms, cone voltage 8 V, and collision energy 20 eV) and *m*/*z* 289.11→97.03 (dwell time 15 ms, cone voltage 8 V, and collision energy 22 eV), respectively.

### 4.3. Assay for Metabolic Stability in Human Liver Microsomes and S9 Fractions

To determine Phase I and Phase II metabolic stability of linalool and linalyl acetate, human liver microsomes and S9 fractions were used, respectively. The assay conditions and reaction mixtures were similar to those used in [24]. Testosterone and 7-hydroxycoumarin were used as positive controls for Phase I and Phase II metabolism, respectively. The reaction mixture was primed for 5 min at 37 °C and then linalool (10 μM), linalyl acetate (10 μM), or a positive control (10 μM) was added. Aliquots of 100 μL were collected from reaction mixtures at the predetermined time points of 0, 10, 20, 30, 45, 60, 90, and 120 min and extracted with 200 μL of ice-cold acetonitrile/methanol (50:50) containing trans, trans-Farnesol (500 ng/mL) as the internal standard. After centrifuging for 15 min at 12,000 RPM (4 °C), the supernatants were transferred to LC-MS/MS inserts and analyzed by UPLC-MS. The elimination half-life (t_1/2_) and intrinsic clearance (CL_int’_; mL/min/kg) were calculated as reported previously [24].
t_1/2_ = 0.693/k
where the slope (k) of the line was obtained by plotting Ln % of linalool and linalyl acetate remaining in the reaction mixture versus incubation time. CL_int’_ = (0.693/in vitro t_1/2_) × (mL incubations/mg microsomes) × (45 mg microsomes/ gm liver) × (20 gm liver/kg BW).

### 4.4. CYP Inhibition Assay

The inhibition of catalytic activity of CYP isoforms was determined using CYP3A4 and CYP1A2 Vivid^®^ kits (Invitrogen, Carlsbad, CA, USA) according to the instructions provided with the kits. Briefly, stock solutions of the extracts, pure compounds, and positive controls were serially diluted in methanol (extracts: 50, 16.67, 5.56, 1.85, 0.62, and 0.21 µg/mL, pure compounds: 25, 8.33, 2.78, 0.93, 0.31, and 0.10 µM and positive control: 1, 0.33, 0.11, 0.03, 0.1, and 0.004 µM) and incubated at room temperature for 10 min with cytochrome P-450 recombinant baculosomes, NADP+, and regeneration system in 96-well, black, round bottom plates at room temperature for 10 min. The reaction was started by adding 10 µL of 10X specific fluorescent substrate of CYP3A4 or CYP1A2. After incubating for a specified time, 50 µL of stop reagent (0.5 M Tris base) was added and fluorescence was measured on the Spectramax M5 plate reader at specified excitation and emission wavelengths for each enzyme. The IC50 values were calculated from concentration response curves generated by plotting percent inhibition versus test concentrations [25].

### 4.5. Reporter Gene Assay for PXR Activation 

The increase in PXR activity by test samples was determined in HepG2 cells transiently transfected with pSG5-PXR (25 μg) and PCR5 plasmid DNA (25 μg) by electroporation at 180 V, 1 pulse for 70 msec, as described earlier [26]. In brief, the transfected cells were plated at a density of 50,000 cells per well in 96-well plates. Upon reaching confluency after 24 h, test samples (60, 20, and 6.7 µg/mL for extract and 30, 10, and 3.3 µg/mL for pure compound) and drug controls (10, 3.3, and 1.1 µM) were added at the indicated concentrations. After incubating the cells with the samples for 24 h, the media was removed and 40 μL of luciferase reagent (Promega Corporation, Madison, WI, USA) was added to each well. Luminescence was measured on a Spectramax M5 plate reader (Molecular Devices, Sunnyvale, CA, USA). Fold increase in luciferase activity of the treated cells was calculated in comparison to vehicle-treated cells. 

### 4.6. Reporter Gene Assay for AhR Activation 

A human AhR reporter assay systems (INDIGO Biosciences, State College, PA, USA) was used for determining the activation of AhR. The assay was performed according to the instructions provided by the supplier and the detailed procedure has been described earlier [27]. Briefly, a 200 µL suspension of reporter cells was added to the wells of a 96-well plate. Cells were allowed to attach for 5 h in a cell culture incubator. After incubation, the media was removed and 200 µL of serially diluted samples were added. The extracts were tested at 30, 10, and 3.33 µg/mL and pure compounds were tested at 10, 3.33, and 1.11 µg/mL. The cells were incubated with the test samples for 24 h. At the end of the incubation period, the medium was removed and 100 µL of luciferase detection reagent was added and after waiting for 5 min at room temperature, luminescence was measured on a Spectramax M5 plate reader. Fold increase in luciferase activity of the treated cells was calculated in comparison to the vehicle-treated cells. Me-Bio was included as positive control.

## 5. Conclusions

In conclusion, we report the in vitro metabolism of linalool and linalyl acetate as well as the effect of LEO, linalool, and linalyl acetate on CYP3A4 and CYP1A2 enzyme activities, PXR and AhR, the two key transcription factors responsible for CYP-450 induction. Our data suggest that rapid in vitro metabolism of linalool and linalyl acetate would limit the oral bioavailability of these compounds. Lastly, based on our results it appears that LEO and its major phytoactives have a low potential for interaction with CYP-450 and, by extension, pose a low risk of drug interactions. 

## Figures and Tables

**Figure 1 molecules-28-00755-f001:**
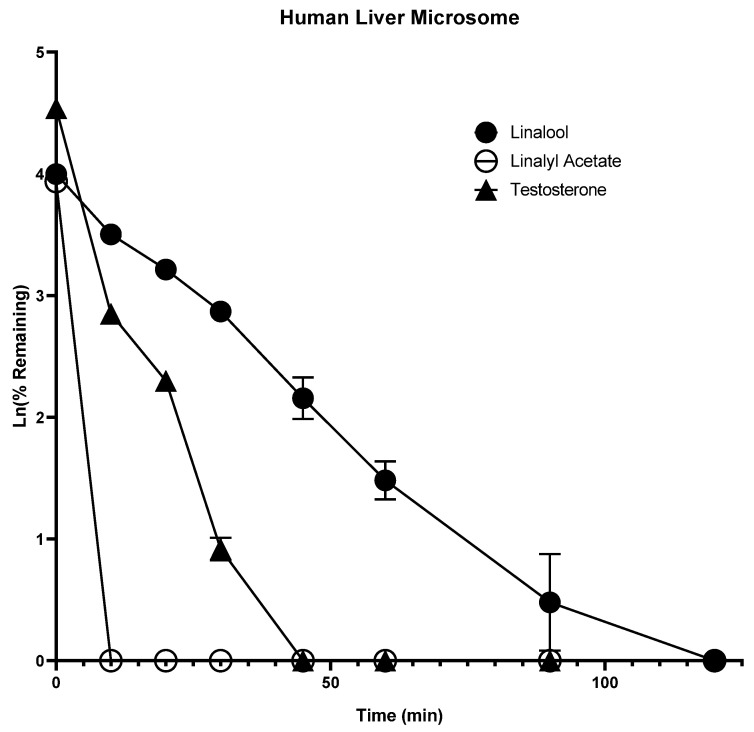
Time-dependent metabolic depletion of linalool, linalyl acetate, and testosterone in pooled human liver microsomes. Human liver microsomes were incubated with linalool, linalyl acetate, and testosterone (10 µM) for 0–120 min in the presence of cofactors. The data are expressed as mean ± SEM.

**Figure 2 molecules-28-00755-f002:**
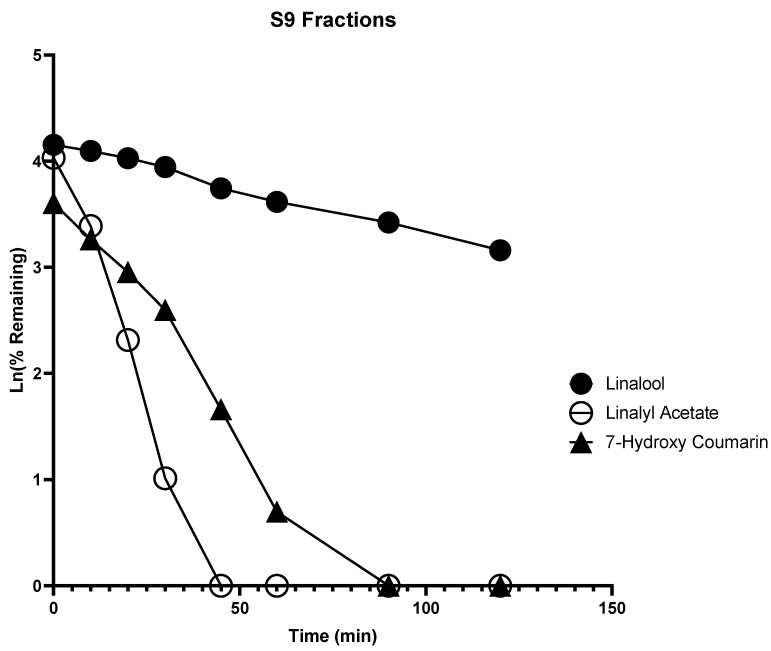
Time-dependent metabolic depletion of linalool, linalyl acetate, and 7-hydroxy coumarin in human S9 fractions. Human S9 fractions were incubated with linalool, linalyl acetate, and 7-hydroxy coumarin (10 µM) for 0–120 min in the presence of cofactors. The data are expressed as mean ± SEM.

**Figure 3 molecules-28-00755-f003:**
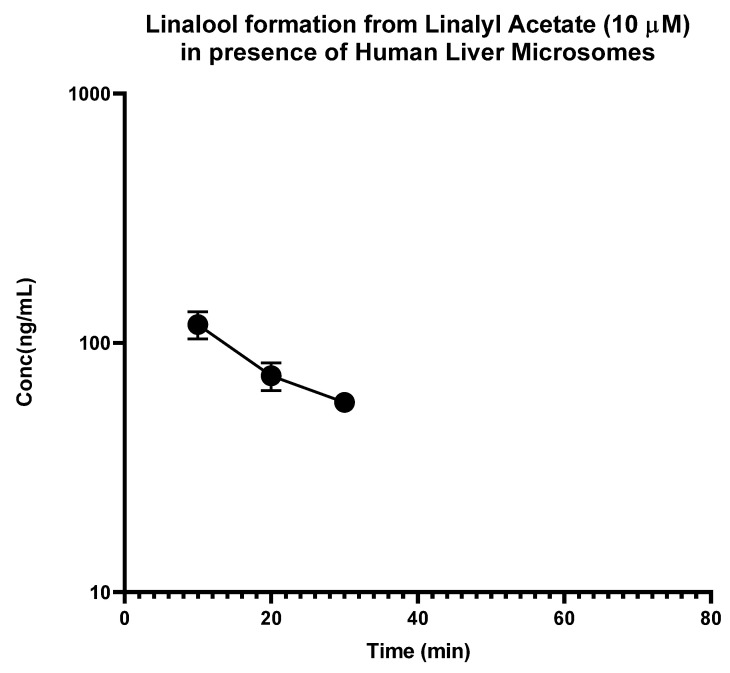
Time-dependent formation of linalool from linalyl acetate in human liver microsomes. Human liver microsomes were incubated with linalyl acetate (10 µM) for 0–120 min in the presence of cofactors. The data are expressed as mean ±SEM.

**Figure 4 molecules-28-00755-f004:**
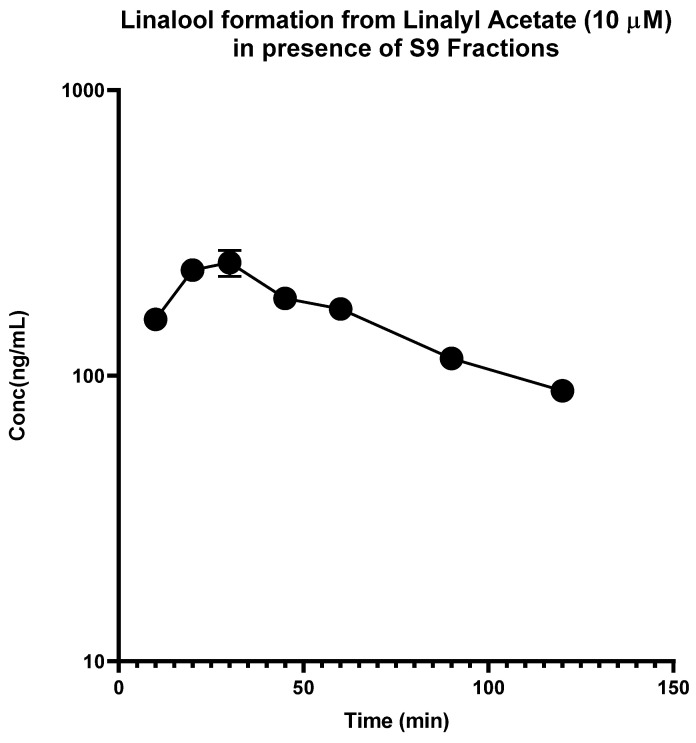
Time-dependent formation of linalool from linalyl acetate in human S9 fractions. Human S9 fractions were incubated with linalyl acetate (10 µM) for 0–120 min in the presence of cofactors. The data are expressed as mean ± SEM.

**Table 1 molecules-28-00755-t001:** IC50 values for inhibition of CYP3A4 and CYP1A2 catalytic activity. The data are represented as mean ± SD. NI = no inhibition.

Compound Name	IC_50_ (µg/mL)	
	CYP3A4	CYP1A2
Lavender oil	12.0 ± 3.00	21.5 ± 0.50
Linalyl acetate	4.75 ± 0.25	NI
Linalool	NI	NI
Ketoconazole * (µM)	0.05 ± 0.01	
α-naphthoflavone * (µM)		0.04 ± 0.01

***** positive control.

**Table 2 molecules-28-00755-t002:** Activation of PXR by lavender oil and its constituents in hepatic cell line (HepG2 cells). Data are represented as mean ± SD.

Compound Name	Concentration	Fold Increase in PXR Activity
Lavender oil	60 µg/mL	1.73 ± 0.14
	20 µg/mL	1.71 ± 0.15
	6.7 µg/mL	1.43 ± 0.23
Linalyl acetate	30 µg/mL	1.83 ± 0.22
	10 µg/mL	1.63 ± 0.18
	3.3 µg/mL	1.45 ± 0.25
Linalool	30 µg/mL	1.67 ± 0.07
	10 µg/mL	1.34 ± 0.08
	3.3 µg/mL	1.33 ± 0.12
Rifampicin *	10 µM	2.82 ± 0.00
	3.3 µM	2.21 ± 0.04
	1.1 µM	1.63 ± 0.17

* positive control.

## Data Availability

The data presented in this study are available on request from the corresponding author.

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
