# Peer review of "In Vitro Metabolism and CYP-Modulating Activity of Lavender Oil and Its Major Constituents"

_molecules, 2023, doi:10.3390/molecules28020755_

Round 1
Reviewer 1 Report
The manuscript “In vitro metabolism and CYP modulating activity of lavender oil and its major constituents” is interesting and original. The authors worked with a current topic of interest to the scientific community, after further reviews I believe that the work can be accepted for publication.
Specific comments:
- Has lavender essential oil been extracted or obtained commercially?
- Has the chemical composition of the essential oil not been analyzed? Why?
- The discussion of the results is superficial. The results should be duly discussed and compared with other studies in the literature.
- What are the study's conclusions?
Author Response
Reviewer #1
The manuscript “In vitro metabolism and CYP modulating activity of lavender oil and its major constituents” is interesting and original. The authors worked with a current topic of interest to the scientific community, after further reviews I believe that the work can be accepted for publication.
Specific comments:
- Has lavender essential oil been extracted or obtained commercially?
Response: The reviewer correctly notes that we failed to reference the previous publication detailing the lavender essential oil characterization. The “Reagents” section has been modified to reflect the fact (Lines 294-295).
- Has the chemical composition of the essential oil not been analyzed? Why?
Response: See previous comment regarding LEO characterization.
- The discussion of the results is superficial. The results should be duly discussed and compared with other studies in the literature.
Response: We note the reviewer’s comment and would like to point out that there are no other studies reporting similar results, ie, this is the first report of in vitro metabolism and P450 modulating activity of LEO and its main constituents as noted in the Discussion.
- What are the study's conclusions?
Response: We thank the reviewer for catching our oversight. Conclusions have been added (Lines 352-364).
Reviewer 2 Report
Although the paper is well written and offers sufficient details, it has no conclusion. I think that research should be continued in order to obtain more information
Author Response
Reviewer #2
Although the paper is well written and offers sufficient details, it has no conclusion. I think that research should be continued in order to obtain more information
Response: Please see response to Reviewer #1 related to Conclusion section
Reviewer 3 Report
This is an interesting work which focus on the herb-drug interaction potential of lavender essential oil, linalool and linalyl acetate. In this manuscript, the authors determined the metabolic stability and inhibitory effect on CYP family of linalool and linalyl acetate. However, I have some questions regarding the experimental discussions, which should be properly addressed prior to publication.
1. It is quite confused about the method of how to obtain S9 fraction ?
2. The relationship of S9 fractions with essential oil, linalool and linalyl acetate is also confused. It seems that the main compounds are linalool and linalyl acetate, but it lacks the corresponding experiments, such as LC-MS/MS.
3. The title, abstract and introduction should be modified further. It is better to make the text seem unified rather than fragmented.
Author Response
Reviewer #3
This is an interesting work which focus on the herb-drug interaction potential of lavender essential oil, linalool and linalyl acetate. In this manuscript, the authors determined the metabolic stability and inhibitory effect on CYP family of linalool and linalyl acetate. However, I have some questions regarding the experimental discussions, which should be properly addressed prior to publication.
- It is quite confused about the method of how to obtain S9 fraction?
Response: The authors note the reviewer’s comment and point out that the S9 fractions were obtained from commercial sources and the procedure to obtain said fraction is not routinely reported as it is a commonly accepted method.
- The relationship of S9 fractions with essential oil, linalool and linalyl acetate is also confused. It seems that the main compounds are linalool and linalyl acetate, but it lacks the corresponding experiments, such as LC-MS/MS.
Response: The reviewer correctly notes that we failed to describe the methodology for measuring linalool and linalyl acetate as well as the control compounds in describing our in vitro metabolism studies. A brief description of the methodology has been added. A detailed description of the method development and validation is currently under review as part of a separate manuscript.
- The title, abstract and introduction should be modified further. It is better to make the text seem unified rather than fragmented.
Response: The authors have reviewed title, abstract, and Introduction and feel they are sufficiently clear for the reader. No edits were made.
Round 2
Reviewer 1 Report
The authors made all requested corrections. I believe the manuscript can be accepted in its present form.
Reviewer 3 Report
1. Lines 293-295: It is confused that as a mixture, how can an essential oil have a purity?
2. The clearance rate of the two compounds alone is not the same as in the mixture. Is it possible that the concentration is different? So, I think the amount of these two substances in the mixture needs to be given. (Lines 152-161)
Author Response
- Lines 293-295: It is confused that as a mixture, how can an essential oil have a purity? Response: The reviewer points out correctly that the statement is confusing. The confusing sentence has been re-written to clarify that it was the individual compounds whose purity was verified. The text has been modified to clarify (Lines 293-294).
- The clearance rate of the two compounds alone is not the same as in the mixture. Is it possible that the concentration is different? So, I think the amount of these two substances in the mixture needs to be given. (Lines 152-161). Response: The mixture was not used to determine the in vitro metabolism, only the pure compounds (Figures 1 and 2).
- Note: The revised manuscript has been uploaded here.